# *MTHFR* and *VDR* Polymorphisms Improve the Prognostic Value of *MYCN* Status on Overall Survival in Neuroblastoma Patients

**DOI:** 10.3390/ijms21082714

**Published:** 2020-04-14

**Authors:** Gladys G. Olivera, Yania Yáñez, Pablo Gargallo, Luis Sendra, Salvador F. Aliño, Vanessa Segura, Miguel Ángel Sanz, Adela Cañete, Victoria Castel, Jaime Font De Mora, David Hervás, Pablo Berlanga, María José Herrero

**Affiliations:** 1Pharmacogenetics Platform, Instituto de Investigación Sanitaria La Fe, 46026 Valencia, Spain; gladysmjiv@hotmail.com (G.G.O.); maria.jose.herrero@uv.es (M.J.H.); 2Department of Pharmacology, University of Valencia, 46010 Valencia, Spain; 3Clinical and Translational Research in Cancer, Instituto de Investigación Sanitaria La Fe, 46026 Valencia, Spain; yanyez_yan@gva.es (Y.Y.);; 4Pediatric Oncology Unit, Hospital Universitario y Politécnico La Fe, 46026 Valencia, Spaincanyete_ade@gva.es (A.C.);; 5Clinical Pharmacology Unit, Hospital Universitario y Politécnico La Fe, 46026 Valencia, Spain; 6Hematology and Hemotherapy Service, Hospital Universitario y Politécnico La Fe, 46026 Valencia, Spain; 7Data Science, Biostatistics and Bioinformatics Platform, Instituto de Investigación Sanitaria La Fe, 46026 Valencia, Spain; 8Department of Pediatric and Adolescent Oncology, Institute Gustave Roussy Center, 94800 Villejuif, France; PABLO.BERLANGA@gustaveroussy.fr

**Keywords:** pharmacogenetics, neuroblastoma, survival, toxicity, SNP

## Abstract

Single nucleotide polymorphisms (SNPs) in Pharmacogenetics can play an important role in the outcomes of the chemotherapy treatment in Neuroblastoma, helping doctors maximize efficacy and minimize toxicity. Employing AgenaBioscience MassArray, 96 SNPs were genotyped in 95 patients looking for associations of SNP with response to induction therapy (RIT) and grade 3–4 toxicities, in High Risk patients. Associations of SNPs with overall (OS) and event-free (EFS) survival in the whole cohort were also explored. Cox and logistic regression models with Elastic net penalty were employed. Association with grade 3–4 gastrointestinal and infectious toxicities was found for 8 different SNPs. Better RIT was correlated with rs726501 AG, rs3740066 GG, rs2010963 GG and rs1143684 TT (OR = 2.87, 1.79, 1.23, 1.14, respectively). EFS was affected by rs2032582, rs4880, rs3814058, rs45511401, rs1544410 and rs6539870. OS was influenced by rs 1801133, rs7186128 and rs1544410. Remarkably, rs1801133 in *MTHFR* (*p* = 0.02) and rs1544410 in *VDR* (*p* = 0.006) also added an important predictive value for OS to the *MYCN* status, with a more accurate substratification of the patients. Although validation studies in independent cohorts will be required, the data obtained supports the utility of Pharmacogenetics for predicting Neuroblastoma treatment outcomes.

## 1. Introduction

Neuroblastoma (NB) is the most common solid extracranial malignancy during childhood with a mean age of diagnosis at 17 months. This type of cancer has its origin in the adrenal medulla or paraspinal ganglia (sympathetic nervous system) during the period of development [1]. It is a disease with great phenotypic heterogeneity, some infants have spontaneous regression of the tumor, while others present disease progression event after intensive multimodal treatment [2].

Familial NB occurs just in 1–2% of cases, with autosomal dominant inheritance. However, practically 99% of NBs are sporadic cases [1]. The first somatic genomic alteration discovered was the amplification of the gene coding for the transcription factor MYCN (proto-oncogene, bHLH transcription factor) on chromosome 2p24, this generally occurs in 20% of cases. This amplification is associated with an aggressive tumor and poorer prognosis [3].

The International Neuroblastoma Risk Group (INRG) has developed a consensus to stratify patients at the time of diagnosis before treatment defining; very low, low, intermediate and high risk categories. Based on clinical data, age, histological type and biological features of the tumor, NBs are classified as Localized (L1, L2), Metastatic (M), or MS (Metastatic Special), in which metastases are confined to the skin, liver and/or bone marrow in children under 18 months [4]. High Risk tumors are defined by the presence of metastasis, but can be found under any of these categories, as long as they have *MYCN* amplification.

Patients with high-risk NB are currently treated in Europe according to the HRNBL1 SIOPEN protocol recommendations, consisting of an induction chemotherapy scheme, “Rapid COJEC”, followed by consolidations with high dose chemotherapy, local treatment (surgery-radiotherapy) and maintenance treatment with retinoic acid and dinutuximab beta [5]. However, the prognosis remains poor, with long-term survival < 50%. New approaches for improving the stratification of patients at initial diagnosis are of capital importance [6]. Here is where Pharmacogenetics (PGx), one of the cornerstones of Personalised Medicine, could become a great value: The analysis of genetic variants, mainly constituted of single nucleotide polymorphisms (SNPs), that make patients respond differentially to drugs in terms of efficacy and toxicity, could propose new markers to add to the existing ones for improved tailored therapy.

Rapid COJEC chemotherapy is composed of Cisplatin (C), Vincristine (O), Carboplatin (J), Etoposide (E) and Cyclophosphamide (C) [7]. Based on this regimen, the dose intensity can be increased in an attempt to achieve an effective treatment with better event-free survival (EFS) and overall survival (OS) [8,9]. All these chemotherapies hold the risk of Adverse Drug Reaction events (ADR), such as infectious complications, gastrointestinal toxic effects, renal toxicity and ototoxicity [10]. ADRs are associated with the genetic makeup of each individual, so personalised and precise medicine approaches are required. The literature has already shown works dealing with PGx, which explain the toxicity observed when treating children with chemotherapy drugs [10,11,12].

In this study, we aim to analyze the impact of chemotherapy-related SNPs on the efficacy and toxicity of NB treatment. With this aim, we performed a PGx study on a retrospective collection of NB patients’ samples (*n* = 95) and associated clinical data. The selection of SNPs to analyze (*n* = 96) was performed by a deep review of the related literature and the information provided in PharmGKB database (www.pharmgkb.org) [13,14] regarding the most commonly employed chemotherapy drugs in NB. In this free-access database, practically all PGx information is compiled and constantly updated. It was created, managed and curated by the University of Stanford and funded by US National Institutes of Health (NIH/NIGMS). Data in this website are under a Creative Commons license. Its goal is the dissemination of knowledge about the impact of human genetic variation on drug responses and on the translation of PGx into clinical practice [15].

Once the samples were successfully genotyped, the frequencies of the different variants for each SNP were summarized and compared to the 1000 Genomes database (www.ensembl.org, 1000 Genomes Project Phase 3) [16], to check for differences in our population compared to the Iberian one in the public database. After that, the genotyping results were correlated to clinical data obtained by retrospective review of Medical Records: (a) OS and EFS outcomes were checked for the whole cohort (*n* = 95); (b) in a subset of patients with high-risk NB treated with Rapid COJEC, efficacy of the treatment in terms of Response to Induction Therapy (RIT) (*n* = 41) was analyzed; (c) in the same group, but only with those patients with the Rapid COJEC completely administered in our Hospital (*n* = 35), chemotherapy-associated severe toxicities were collected (grades 3–4, according to Common Terminology Criteria for Adverse Events, CTCAE, v4.03).

The main results found: (1) Frequencies that are different from those expected in 19 SNPs; (2) 8 SNPs related with severe or life-threatening chemotherapy-associated toxicities; (3) SNPs influencing RIT in patients with high-risk NB and (4) SNPs in *MTHFR*, *VDR* and *SOD2* genes identified as prognostic factors of OS or EFS, independently of *MYCN* amplification, underlying differential survival prediction possibilities independent of *MYCN* status.

Although promising, these results need to be validated in larger independent cohorts.

## 2. Results

Starting from the initial 100 NB samples, 5 were discarded because they did not fulfill the necessary quality or quantity for the DNA extraction and genotyping methods. Therefore, 95 samples from 95 NB patients remained for this work analyses.

Table 1 shows the basal characteristics of the patients included.

Figure 1 shows the number of samples included in the whole study, indicating the subsets of samples that have been chosen for the different analyses reported in Results section. From the total amount of 100 newly diagnosed frozen samples that were available in the retrospective collection, 5 were not suitable for genotyping, and 95 remained for use. Those 95 made up the entire cohort. Each of the following paragraphs in this section explains the criteria for choosing the subgroups. In brief, the SNP variants frequency analyses, OS and EFS studies were performed on the whole cohort. From those 95 samples, only 41 were High Risk NB treated with Rapid COJEC and on those, RIT relation to SNPs analysis was analyzed. Inside that group of 41 patients, 35 completed the whole Induction treatment in the same hospital, so analysis of chemotherapy-toxicity related to SNP could be performed.

### 2.1. Genotyping and Comparison of Observed Variant Frequencies with 1000 Genomes Data

After genotyping the 96 SNPs panel in the 95 samples, in order to check the quality of the results and the similarity with described populations in international databases, the frequencies obtained for each SNP variants were compared to the expected ones in 1000 Genomes database Project Phase 3, at the IBS population (Iberian Populations in Spain, the most accurate for our cohort). Statistically significant differences were found for 19 SNPs in 18 different genes. The analysis was performed by Chi squared test (with False Discovery Rate penalization), considering significant *p*-value < 0.05. Table A1 in Appendix A shows the results of those SNPs with statistically significant different frequencies.

### 2.2. SNPs Related to the Toxicity Caused by Chemotherapeutic Drugs During Induction Treatment

In order to look for associations between SNPs and chemotherapy toxicity, the most homogeneously treated group of patients was selected: High Risk, all treated with the same induction therapy, Rapid COJEC, and with the whole treatment completed at our Hospital, in order for the clinical records to be complete and homogeneously recorded during that induction. 35 patients fulfilled these conditions. The elastic net statistical model selected two SNPs, rs2228001 in *XPC* and rs7779029 in *SEMA3C*, predictive of severe gastrointestinal toxicity. Six more SNPs were identified as predictive of serious infectious complications: rs1799793 in *ERCC2*, rs1801133 in *MTHFR*, rs2306283 in *SLCO1B1*, rs2784917 in *SLIT1*, rs6907567 in *SLC22A16* and rs7186128 which is not located in any identified gene.

### 2.3. SNPs Related to Response to Induction Therapy

In our aim to find associations between SNPs and efficacy of the treatment, a first analysis was performed evaluating the efficacy of the induction chemotherapy. After collecting the data of RIT in High Risk-Rapid COJEC treated patients (*n* = 41), their response was categorized as metastatic Complete Response or non-metastatic Complete Response. The SNPs associated with RIT according to Logistic Regression models penalized with Elastic Net (covariates *MYCN* status, patient age, tumor stage) are shown in Table 2.

### 2.4. SNPs Related to OS and EFS

Our final goal was to find associations between the SNPs and the long-term efficacy of the treatment, evaluating survival data from the whole cohort of patients (*n* = 95). This was tested with both OS and EFS. The associations found employing Cox regression models penalized with Elastic Net (covariates MYCN status, patient age, tumor stage) are shown in Table 3.

Based on these findings, and the knowledge that the *MYCN* status is one of the most reliable parameters for risk prediction in NB, we decided to confirm the effect of *MYCN* status in our series of patients. As expected, patients with amplified *MYCN* tumors (*n* = 29) only reach a 20% rate in OS (Kaplan-Meier, continuous follow-up and censored data), while when *MYCN* is not amplified (*n* = 66) the OS rate was 50% approximately (Figure A1).

Based on the aim of going on a step further, we decided to check whether the combination of these new findings (Table 3) with *MYCN* status could improve the identification of patients with better, or poorer, prognosis in the survival curves. The results are shown in Figure 2 for OS and Figure 3 for EFS. In these two figures, the analyses test whether there are differences in the survival curves between the effect of *MYCN* status on its own (grey lines), compared with the effect of the corresponding *MYCN* status, but dividing the patients according to their genotype for each SNP (blue or red lines). Significant p-values would mean that the stratification adding the genotype to MYCN status is statistically different from MYCN status alone, and the p-value will be particularly related to the most different variant comparing its colored line to the grey line.

From the three selected SNPs related to OS, rs1801133 at *MTHFR* and rs1544410 at *VDR*, showed statistically significant results between the different possible variants and *MYCN* status alone (*p* = 0.02 for TC genotype, and *p* = 0.006 for GA genotype, respectively). The other, rs7186128, as Figure 2b shows, offers no additional significant value to *MYCN* status information, although GG patients’ curves are clearly separated from their grey global curves.

With respect to EFS, the same analyses as for OS were performed. The results are shown in Figure 3. Two of the selected SNPs, rs45511401 in *ABCC1* and rs3814058 in *NR1I2* could not be included in this type of analysis due to the very low number of patients in two of the three possible categories (variants). Statistically significant results were found for rs1544410 (GA, *p* = 0.027 compared to the grey line), rs65398770 (GG, *p* = 0.031 compared to the grey line) and rs4880 (TC, *p* < 0.001 compared to the grey line), but not for rs2032582.

## 3. Discussion

There is an unmet need to improve current treatment and survival of patients with NB. While, there are many groups working on the study of tumor genetic variants related to the prognosis and efficacy of the treatment, this is not the case in relation to the analyses of constitutive genetic variants that could help to improve the personalised precision treatment of these tumors, under the discipline of PGx. The present study focuses on the analyses of a set of SNPs (96), all of them related to the efficacy and toxicity outcomes of chemotherapeutic drugs employed for NB treatment: Mainly in Rapid COJEC induction treatment (cisplatin, carboplatin, cyclophosphamide, etoposide and vincristine), but also including SNPs, related to drugs that may be administered later, after induction treatment (as TVD: Topotecan, vincristine, doxorubicin). The aim is to find better ways to predict which patients will be “good or bad responders” to the treatment. If we employ preemptive PGx testing for this, we would be able to identify these patients at the earliest possible and adjust their treatment in a personalised way.

In order to perform the selection of SNPs to be included in the study panel, as well as in the interpreting the results, PharmGKB (together with the published literature) was the main resource, as it is widely considered the most relevant provider of PGx information [13,14,15]. It counts with a group of experts working on the dissemination of knowledge about the impact of human genetic variation on drug responses and on the translation of PGx into clinical practice. It is mainly based on the results of PGx articles, published worldwide, and the majority are included in PubMed database. PharmGKB is not only a compilation of published articles, it also curates and classifies the information extracted from them by its experts, and gives back their evaluation of the evidence provided by each work. In this manner, they classify drug/polymorphism relationships under the name “Clinical Annotations”, graded with a “Level of Evidence” label. This scientific evidence rank is assigned from 1 to 4 (1A, 1B, 2A, 2B, 3 and 4), 1A being the highest. In level 1A annotations, the variant-drug combination is included in a Clinical Pharmacogenetics Implementation Consortium (CPIC) guideline or a PGx guide approved by a medical society or implemented in a PGRN (Pharmacogenomics Research Network) site or in another important health system. In level 1B annotations, the preponderance of evidence shows an association that replicates in more than one cohort with significant p-values, and preferably with a strong effect size. Level 2 includes variants with moderate evidence. Level 2A marks annotations for variant-drug combinations that qualify for level 2B where the variant is within a VIP (Very Important Pharmacogene), as defined by PharmGKB, so functional significance is more likely. In Level 2B variant-drug combinations, the association must be replicated but there may be some studies that do not show statistical significance, and/or the effect size may be small. The following two Levels are less strongly supported. Level 3 annotations are based on a single significant (not yet replicated) study or variant-drug combinations, evaluated in multiple studies, but lacking clear evidence of an association; and finally, level 4 annotations are based on a case report, non-significant study or in vitro, molecular or functional assay evidence only. The assignment of variant-drug Levels of Evidence, mainly in cancer-related drugs and especially at Levels 3 and 4 can lead to misinterpretations if the reader is not an expert in the field.

The main problem in cancer is that most of the PGx studies in cohorts of patients are observational ones and performed under complex therapeutic schemes. Therefore, assigning the observed effect of a SNP with a unique drug is difficult, nearly impossible, with the usual study designs. As a consequence, when SNPs effects are found, and a chemotherapy scheme with several combined drugs has been administered, the trend is giving the responsibility of the effect to the drug where the gene containing that SNP is supposed to have an effect. However, that gene could in fact be related to other drugs of the scheme, but their relationship not known yet or poorly described. It is also important to keep in mind, that in spite of being using chemotherapeutic drugs for decades, for many of them we still do not know completely all the genes coding for their transporter proteins, metabolizing enzymes or mechanism of action-related molecules. Although our panel was designed in 2014, all the selected SNPs are still suitable, as we checked recently (PharmGKB last accessed January 2020). Nevertheless, it is true that this field evolves very quickly, and apart from new variants appearing in the published studies, the already described ones can change in their Level of Evidence, as more data sets appear, they are curated and do influence this parameter.

Before discussing the concrete results, it is relevant to keep in mind that the statistical approaches required in this type of studies need to be advanced and complex ones, able to evaluate sets of data where there are more variables than observations (patients). In fact these analyses are statistical models instead of statistical tests, and usually do not report the results in the “classical statistics” manner. For instance, as described and referenced in Materials and Methods section, some of them do not give a *p*-value as a result or a 95% Interval of Confidence as an estimate for a given parameter. Another relevant concern to assess before starting analyzing toxicity or efficacy was evaluating if there were SNPs in our panel that could be directly linked to a higher probability for a patient to be High Risk. In our whole cohort, 75 patients were High Risk and 20 non-High Risk. Only GG patients at SNP rs1979277 showed lower basal risk of being High Risk NB (*p* = 0.016, Elastic Net logistic regression). This SNP is not part of any of the following results that correlate SNPs with efficacy or toxicity of the treatment, so we can conclude that no interference is caused. From another perspective, trying to find out which could be the explanation of this finding, the underlying mechanisms and the potential utility could be of interest for future researches.

### 3.1. SNP Variants Differentially Represented

The comparative analysis of the frequencies of each SNP obtained in our cohort with respect to those expected according to the 1000 Genomes Project Phase 3 database reveals statistically significant differences in 19 polymorphisms, from the total of 96 that were evaluated. These differences may be due to real population differences between our cohort, where 90% of the patients are from the Valencian Community, compared to the rest of the Spanish population (IBN). On the contrary, they could be due to the fact that the significant variants could be intrinsically related to the tumor process, in contrast with healthy individuals of a comparable population. To solve this question, future studies of the same set of SNPs, comparing with a healthy cohort of individuals from the Valencian Community are required. Any of the two possible results are interesting. If the differences are not related to the disease process, but are specific of Valencian population, these data could be of clinical interest as some of the SNPs with frequencies different from the expected have been found to cause clinical effects in the present and other works [10,11,12,15]. Understanding in advance that patients bearing these variants can be found with higher, or lower, frequency in a given population could make clinicians aware, in advance, of the possible occurrence of their clinical effects. On the contrary, if the differences in frequency were proved to be linked to the tumor process, those SNPs could become NB predisposition-associated markers, once validated in larger comparable cohorts.

### 3.2. Toxicity Analysis

For this substudy, only those High Risk patients in our cohort that completed their whole Rapid COJEC induction treatment in our hospital were included, in order to avoid incomplete information and heterogeneity in reporting toxicity results that could lead to erroneous conclusions. Only grade 3 and 4 toxicities, according to CTCAE v4.03, were collected for SNP correlation analyses. As expected, having a group of 35 patients, most of the types of toxicity collected could not be included in the analyses because there were not enough grade 3–4 observations directly attributable to chemotherapy. However, regarding gastrointestinal and infectious events, we were able to perform the SNP correlation analyses employing Empirical Bayesian Elastic Net methods. This method selected two SNPs related to gastrointestinal toxicity: rs2228001 in *XPC* gene, Level 1B at PharmGKB, found associated in the literature with cisplatin toxicity; and rs7779029 in *SEMA3C* gene, that has been related in the literature with NB dissemination [17]. Regarding infectious complications, the model selected 6 SNPs: rs1799793 in *ERCC2* gene, Level 3, described in the literature to be associated with toxicity of platinum-compounds and etoposide, producing anemia and pneumonitis. Rs1801133 in *MTHFR*, Level 2A, related with methotrexate but also platinum-compounds toxicity. Rs2306283 in *SLCO1B1*, Level 3, was described to be related with irinotecan and methotrexate, but also other SNPs in this gene are related to cyclophosphamide and etoposide transport into the cells. Rs2784917 in *SLIT1*, Level 4, was related to etoposide toxicity. Rs 6907567 in *SLC22A16*, Level 3, related with cyclophosphamide and doxorubicin toxicity and efficacy. And finally, rs7186128 not located in described genes, Level 3, related to cisplatin and irinotecan efficacy.

If these results could be validated in different cohorts, the set of SNPs could become a predictive signature for identifying patients at risk of developing these toxicities.

### 3.3. Response to Rapid COJEC Induction Therapy

All 41patients in our cohort, that were classified as High Risk and were treated with Rapid COJEC, were evaluated as achieving mCR or non mCR after their induction therapy. The correlation with the SNPs panel was performed with Logistic Regression models with Elastic Net penalty method, including as covariates *MYCN* amplification status, age and stage. The model selected 4 SNP variants associated with Odds Ratios larger than 1, indicating more probability of response to treatment. Two of them are associated in the literature, having efficacy of platinum-compounds: rs3740066 GG variant in *ABCC2* (related to platinum-compounds efflux out of cells and Level 3, toxicity of cyclophosphamide, irinotecan and doxorubicin) and rs726501 AG variant in *MAP3K1* (Level 3, efficacy of platinum-compounds and taxanes). The other two related in the literature with cyclophosphamide efficacy: rs1143684 TT in *NQO2* (in agreement with Level 3 [18]) and rs2010963 GG in *VEGFA* (in agreement with Level 3, [19]).

On the other hand, there were 4 SNP variants associated with Odds Ratios lower than 1, indicating less probability of response to treatment. Three of them were already associated in the literature with cyclophosphamide efficacy. Rs10276036 TT in *ABCB1* (Level 3, one publication evaluating another efficacy parameter, being TT associated with decreased risk of death when treated with cisplatin, cyclophosphamide, doxorubicin, methotrexate, and vincristine [20]). In *SLCO1B1*, rs4149056 TC has not been related yet to efficacy of treatment in the literature, but generally to decreased toxicity instead (Level 3, several chemotherapy different drugs) [21]. In *CBR3*, rs8133052 GG, the main findings have been published in relation to doxorubicin [22], but other SNPs of this gene have been related with cyclophosphamide (Level 3, toxicity) [23]. The fourth variant, rs1544410 GA, is located in *VDR*, the gene coding for Vitamin D receptor, also named *NR1I1*. This variant showed to confer less activity to the VDR protein [24], which is involved in the regulation of *CYP3A4* [25], one of the main metabolizing enzymes (together with *CYP3A5*) of etoposide. We hypothesize that this could be the link for the effects observed: lower levels of VDR could results in lower levels of etoposide metabolism and therefore, lower RIT.

### 3.4. Efficacy in Terms of Survival

These set of analyses are the ones, together with frequency analyses, that include the whole cohort of 95 NB patients. Employing Cox regression models, penalized with Elastic Net and including as covariates again the main prognostic factors in NB, *MYCN* status (normal or amplified), and patient’s age and tumor stage. Regarding OS, 2 SNPs variants were found predictive of better survival probability (Hazar Ratio < 1) and 1 SNP predictive of lower survival probability (Hazard Ratio > 1). On the other side, for EFS, 3 SNPs variants were selected as better survival predictors, and other 3 as lower survival ones. At this point, in order to analyze if the SNPs could impact and add new value to the “gold standard” prognosis factor, we decided to add these selected SNPs, to the corresponding OS or EFS curves of the whole cohort, according to *MYCN* status. Figure 2 and Figure 3 show the results.

Regarding EFS, three SNPs, rs6539870, rs1544410 in *VDR* and rs4880 in *SOD2* provided a further patients’ sub-stratification compared to *MYCN* status alone. In this sense, rs1544410 in *VDR,* with variant curves different (*p* = 0.027) from the global one (*MYCN* status alone), shows that in Normal *MYCN* tumors, GG patients present a very much different prognosis, achieving long-term 50% EFS, instead of the global long-term 20%, according to *MYCN* only. On the contrary, GA patients show worse results, with 10% EFS shortly after 100 months of follow-up, and failing to 0% shortly after 200 months. VDR possible link to NB therapy was already discussed in the previous paragraph. The SNP rs6539870 has been described in a Genome Wide Association Study as related to etoposide toxicity [26], with GG variant, providing higher risk of toxicity. This variant is in fact the one that confers worse EFS results to our patients in Figure 3c. The last SNP, rs4880 in *SOD2* presents the most significant *p*-value (0.001). It is related with cyclophosphamide efficacy, with a 2A Level of Evidence in PharmGKB, describing that TT patients achieve longer survival than CC in breast cancer [27], which is also true in our study. We further observe, that GG patients show an infaust prognosis in both, *MYCN* amplified or not, scenarios.

In the case of OS, two of the three SNPs presented substratifications according to their variants, that were statistically significant compared to the data taking only *MYCN* status into account (grey line). These were rs1801133 in *MTHFR* and rs1544410 in *VDR.* The recurrence of these two SNPs in the whole study results is remarkable. In the case of the first, with a *p*-value = 0.02, Figure 2a, shows how TC patients always present higher OS percentages than the global ones (*MYCN* status alone), while CC, on the contrary, always present worse results, less than 30% OS in normal *MYCN* tumors and around 10% in amplified ones, leaving these patients at higher risk of death. This SNP is Level of Evidence 2A, regarding efficacy of carbo and cisplatin, describing CC patients as those with decreased response [28], but most of the publications are related to methotrexate toxicity [29,30]. *MTHFR* lies at the intersection of the pathways for DNA methylation and DNA synthesis. It catalyzes the reduction of 5,10-methylenetetrahydrofolate to 5-methyltetrahydrofolate, the substrate for conversion of homocysteine to methionine. The most widely studied SNPs in this gene are rs1801133 (also called 677C > T) and rs1801131 (also 1298C > A). Our SNP, rs1801133, was associated with 50% lower activity of the enzyme in vitro [31], and was the first genetic risk factor identified for *Spina Bifida*. A study published in 2016, described a regulatory effect of *MYCN* on the expression of *MTHFR*, repressing it [32], and some time before, in ANR (Advances in Neuroblastoma Research Association) meeting 2014, an abstract by N. Svergun et al. [33] concluded that this SNP was related to NB susceptibility and *MYCN* amplification. We could hypothesize that an increase in *MYCN* expression, could lead to downregulation of *MTHFR* and this subsequently, lead to a decrease in the routes of DNA synthesis and DNA methylation. If this is true, then the variant TT could be boosting the effect in *MYCN* amplified patients even further, since TT has been described as lowering the activity of MTHFR enzyme, and that is what we see in our patients.

Understanding the effect of this SNP in chemotherapy treatments is of great relevance since there is a significant amount of published results, with no clear consensus yet on its effects.

In relation to rs1544410, with a *p*-value = 0.006, when comparing with the global line, Figure 2c, we observed a similar behaviour as in EFS: The results are clearer in normal *MYCN* tumors rather than in amplified ones. This could be suggesting some linkage between the two genes, but further studies with larger number of patients are required to confirm this. In the non-amplified *MYCN* tumors, GG patients reach 70% OS rate, while the global data was 50%, and GA group only reaches 30%. On the other hand, in the *MYCN* amplified group, AA patients achieve 50% OS at approximately 160 months, while the global line stays at 20% for the same time period.

It is noticeable that amongst all the panels in Figure 2 and Figure 3, considering only those with statistically significant p values, the results from *MTHFR* and *VDR* SNPs show apparent differences between normal and amplified *MYCN*. Regarding *MTHFR* and OS, while TC effect seems clear, the effects of TT and CC look opposite between normal or amplified *MYCN* patients. As to *VDR*, in both OS and EFS figures, the effect of GA is maintained in both scenarios, while GG and AA show different results. Unfortunately, we do not have an explanation for these findings. The number of patients in each genotype category could be responsible of these effects. Also, the treatment applied to patients in each scenario, normal versus amplified, could influence these differences: The 66 *MYCN* amplified-patients are High Risk, so more intensively treated, while in the 29 normal *MYCN* patients, 9 of them are High Risk, and 20 are not (we have a total of 75 High Risk patients). Further studies are needed to explain the differences observed in those genotypes.

Being able to predict which patients will respond better to the therapy before starting a treatment [34], in terms of greater efficacy and/or lesser toxicity, is of paramount importance especially in areas such as pediatric oncology. Although the results should be validated in different comparable cohorts, we can conclude that it seems possible to use PGx polymorphisms as predictive elements to stratify patients with different probabilities of therapeutic success, and consequently adapt the therapy.

## 4. Materials and Methods

A total of 100 NB patients were selected for the study, whose peripheral blood samples had been collected and stored frozen, after written informed consent obtention, in the Pediatric Oncology Unit of Hospital La Fe (Valencia, Spain). These patients were diagnosed between 1989 and 2015, and included in different national and European studies (INES, EUNS, N-AR-99, N-II-92, LINES and HR-NBL1). Staging and risk stratification were established according to International Neuroblastoma Risk Group (INRG) criteria [35]. Biological studies including status of *MYCN,* were performed according to ENQUA guidelines [36]. Retrospective collection of clinical data was performed using Medical Records. For chemotherapy toxicity, CTCAE v. 4.03 was employed (standardized definitions for adverse events published by the National Cancer Institute (NCI) of the National Institutes of Health (NIH)).

Ethical approval was obtained from Hospital La Fe Ethics Committee with the project codes PIE13/00046 ISCIII and 2016/0150 Fundación Mutua Madrileña.

This study has been developed from 2014 until 2019, and the design of the SNPs panel was performed in 2014 (MJ.H. and SF.A.), by means of a deep review of the related literature and the information provided in PharmGKB database (www.pharmgkb.org) [12,13] regarding most of the chemotherapy drugs employed in NB. These were: Cisplatin (the searches were amplified to platinum-compounds), vincristine, carboplatin, etoposide, cyclophosphamide, topotecan (searches amplified to irinotecan, because of molecular target similarity) and doxorubicin. The selected SNPs, a total of 96, belonging to 60 different genes, have been revised while preparation of this article, and they are all still valid.

### 4.1. Genotyping and Comparison of Observed Variant Frequencies with 1000 Genomes Data

After processing peripheral blood samples for DNA extraction, 5 samples were not found suitable for further analyses due to poor DNA quality (*n* = 3) and low sample quantity (*n* = 2), and therefore 95 samples from different NB patients remained for the analyses.

Genomic DNA from each sample was isolated from 200 µL of whole blood using a commercially available kit based on centrifugation in microcolumns (UltraCleanBloodSpin DNA Isolation Kit; MoBio Laboratories, Inc., Carlsbad, CA, USA). DNA concentration and purity were measured with a NanoDrop spectrophotometer (NanoDrop Technologies Inc., Wilmington, DE, USA), according to the manufacturer’s instructions, using the provided absorbance at 260nm for the first and the 260/280 ratio for the second. DNA was stored at −20 °C until use for genotyping.

Genotype was performed in triplicates for each sample, with the Sequenom MassArray platform, at UCIM-Faculty of Medicine facilities, University of Valencia. All the SNPs (Table 4) were genotyped according to the manufacturer’s instructions (former Sequenom, currently AgenaBioscience, San Diego, CA, USA) employing 200 ng of DNA from each sample.

To compare the observed variant frequencies with the expected at Phase 3 database, the IBS population (Iberian Populations in Spain), Chi Squared test False Discovery Rate penalization was performed, considering the results statistically significant when *p*-value < 0.05. 1000 genomes Project was last accessed in September 2019, https://www.ensembl.org/Homo_sapiens/Info/Index?db=core [16].

### 4.2. SNPs Related to the Toxicity Caused by Chemotherapeutic Drugs During Induction Treatment

Following the CTCAE v 4.03, toxicities attributable to the chemotherapeutic drugs from 35 High Risk patients were retrospectively retrieved from the Medical Records by two different pediatric oncologists (P.B. and P.G.) and their results were cross-checked (double-blind). In case of discrepancy, the data were discussed and agreement achieved. Only grades 3 and 4 were considered for the analyses as these are more robust and objective, compared to grades 1 and 2.

Selection of the SNPs related to the toxicities was performed using an Empirical Bayesian Elastic Net algorithm [37]. Elastic Net is a variable selection method that builds predictive models including only variables with predictive power. It is especially suited for data with many variables and few observations and it does not produce p-values. Variables included in the model are considered important and variables not included in the model are considered not important. Elastic Net does not calculate Confidence Intervals, it only generates predictive models, not inferential ones.

### 4.3. SNPs Related to Response to Induction Therapy

Our cohort has 53 High Risk NB patients, but only 41 of them received Rapid COJEC as induction therapy, so in order to work with a comparable set of patients, we only studied the associations of the genotypes of this 41 samples subset with evaluation of the efficacy of Induction therapy according to mCR vs. non mCR [38], by means of Logistic Regression models penalized with Elastic Net [39], including as covariates the already described main prognostic factors in NB: *MYCN* amplification status, patient age and tumor stage.

### 4.4. SNPs Related to OS and EFS

The association of genotypes with survival data includes Event-Free Survival (EFS), that is, the period of time from diagnosis until the time of first occurrence of relapse, progression or death. Also Overall Survival (OS), defined as the period of time from the date of diagnosis until death or until last contact if the patient was alive. This analyses were performed with the whole cohort (*n* = 95 patients), with a minimum follow up of 0, 66 and a maximum of 279 months after diagnosis. Cox regression models were employed, penalized with Elastic Net [40] and the already indicated main prognostic factors in NB were included as covariates: *MYCN* status (normal or amplified), patient’s age and tumor stage.

## Figures and Tables

**Figure 1 ijms-21-02714-f001:**
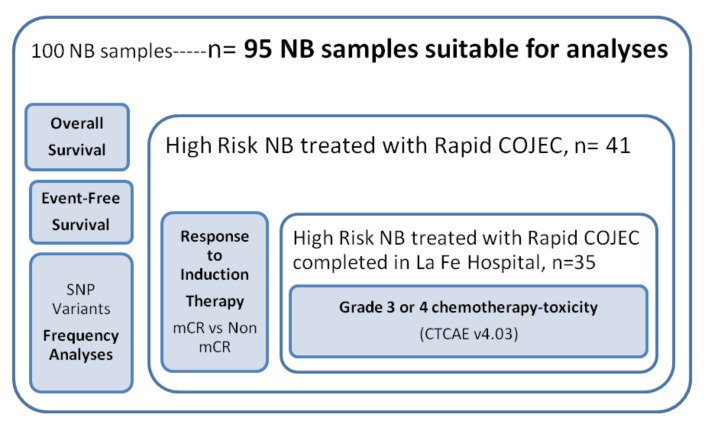
Patients included in the study and analyses performed. CTCAE: Common Terminology Criteria for Adverse Events; mCR: metastatic Complete Response; Non mCR: non metastatic Complete Response; NB: neuroblastoma; SNP: Single Nucleotide Polymorphism.

**Figure 2 ijms-21-02714-f002:**
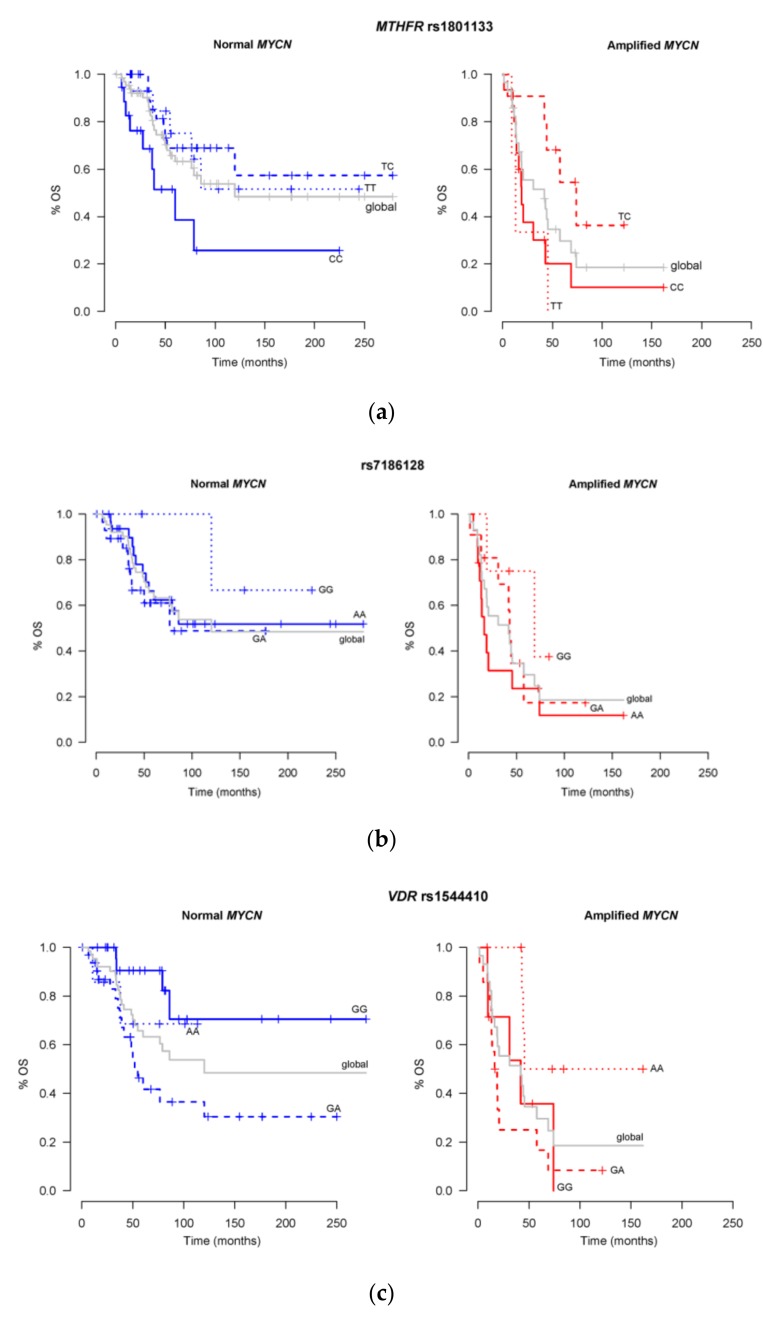
Effect of the addition of the OS related SNPs, selected by Cox regression and Elastic Net, to the OS prognosis effect of *MYCN* status. The effect of *MYCN* status alone (Normal, *n* = 66, on the left panels or Amplified, *n* = 29, on the right panels) is represented by the grey line named “global”, while the same patients’ data distributed according to the SNP variants, are represented by the colored lines. (**a**) Shows the results according to SNP variants in rs1801133 at *MTHFR* gene, with TC genotype achieving *p* = 0.02 statistically significant difference comparing with the grey global lines. (**b**) shows the results according rs7186128 variants, with no statistical significance comparing with the global lines. (**c**) Shows the results according to SNP variants in rs1544410 at *VDR* gene, with GA genotype achieving *p* = 0.006 statistically significant difference comparing with the global grey lines.

**Figure 3 ijms-21-02714-f003:**
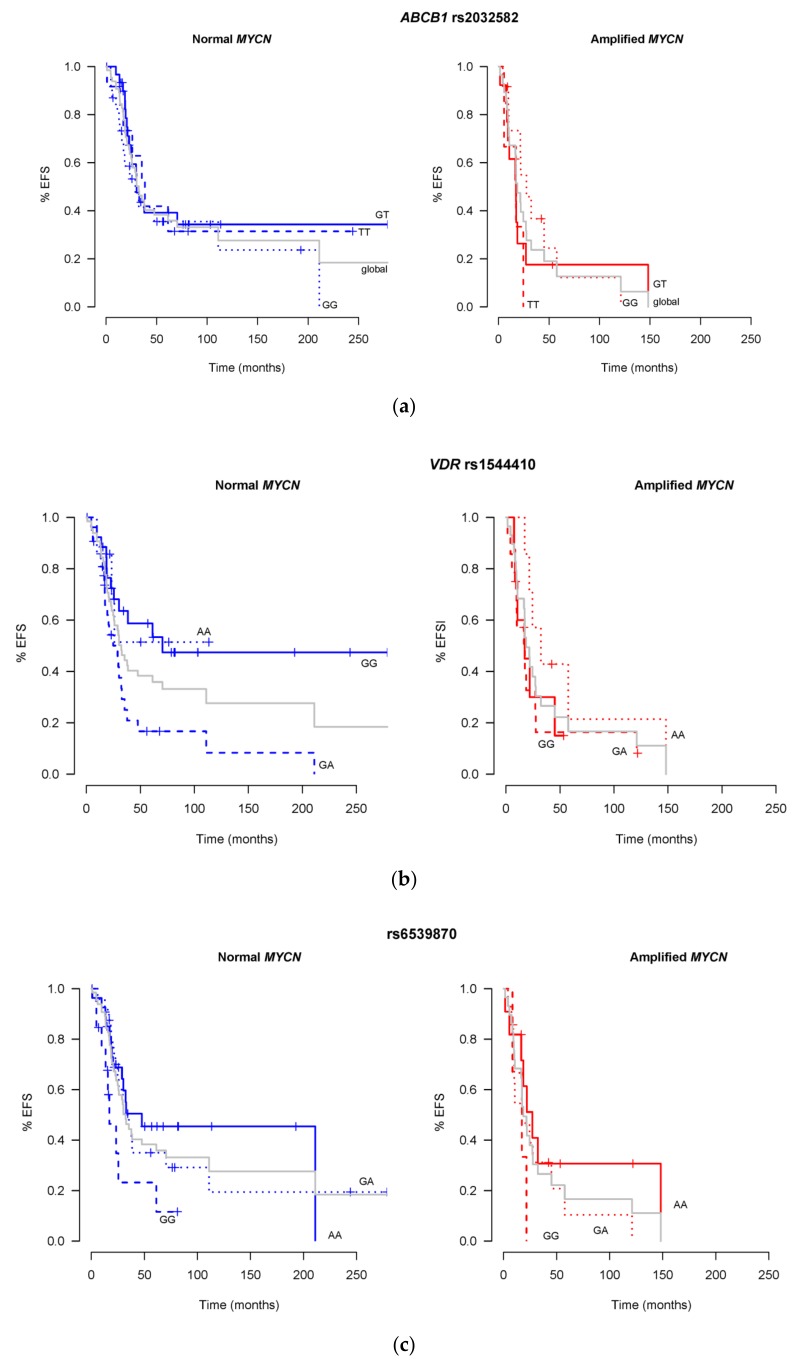
Effect of the addition of the EFS related SNPs, selected by Cox regression and Elastic Net, to the EFS prognosis effect of *MYCN* status. The effect of *MYCN* status alone (Normal, *n* = 66, on the left panels or Amplified, *n* = 29, on the right panels) is represented by the grey line named “global”, while the same patients’ data distributed according to the SNP variants, are represented by the colored lines. (**a**) Shows the results according to SNP variants in rs2032582 at *ABCB1* gene, with no statistical significance comparing with the global lines. (**b**) Shows the results according to rs1544410 variants at *VDR* gene, with *p* = 0.027 for GA genotype, reaching statistically significant difference comparing with the global lines. (**c**) Shows the results according to SNP variants in rs6539870 at a region with no identified gene, with GG genotype reaching p=0.031 statistically significant difference comparing with the global lines. (**d**) Shows the results according to rs4880 variants at *SOD2* gene, with TC reaching *p* < 0.001 statistically significant difference comparing with the global lines.

**Table 1 ijms-21-02714-t001:** Descriptive characteristics of the study patients.

Variable	*n* = 95	Variable	*n* = 95
	Mean (SD)/*n* (%)		Mean (SD)/*n* (%)
	Median (1st, 3rd Q.)		Median (1st, 3rd Q.)
Overall survival (months)		State at last follow-up	
	59.72 (57.78)	Alive	52 (54.74%)
	42.97 (17.77, 77.65)	Exitus	43 (45.26%)
Progression-free survival (months)		Metastasis	
	41.57 (50.58)	None	30 (31.58%)
	22.97 (15.03, 48.97)	Yes	65 (68.42%)
Age at initial diagnosis (months)		*MYCN* status	
	39.55 (37.14)	Amplified	29 (30.53%)
	32.3 (14.6, 49)	Normal	66 (69.47%)
Relapse		INRG Stage	
No	32 (33.68%)	L2	30 (31.58%)
Yes	63 (66.32%)	M	61 (64.21%)
		Ms	4 (4.21%)

International Neuroblastoma Risk Group (INRG), L2: Localized tumor with one or more image-defined risk factors, M: Distant metastatic disease, Ms: Metastatic disease in children under 18 months with metastases limited to skin, liver, and/or bone marrow (<10% involvement).

**Table 2 ijms-21-02714-t002:** SNPs related to Response to Induction Therapy, Rapid COJEC in High-Risk NB.

Gene	SNP	Results			Involved Candidate Drug *
		Genotype	OR	Association	
mCR vs non mCR					
*ABCC2*	rs3740066	GG	1.79	↑ efficacy	Platinum-compounds
*MAP3K1*	rs726501	AG	2.87	↑ efficacy	
*NQO2*	rs1143684	TT	1.14	↑ efficacy	Cyclophosphamide
*VEGFA*	rs2010963	GG	1.23	↑ efficacy	
*ABCB1*	rs10276036	TT	0.67	↓ efficacy	
*SLCO1B1*	rs4149056	TC	0.64	↓ efficacy	
*CBR3*	rs8133052	GG	0.53	↓ efficacy	
*VDR*	rs1544410	GA	0.68	↓ efficacy	Etoposide

mCR: metastatic Complete Response; non mCR: non metastatic Complete Response. OR: Odds Ratio. SNP: Single Nucleotide Polymorphism. * According to literature and PharmGKB. Total number of patients, *N* = 41. ↑ means “increase” and ↓ “decrease”.

**Table 3 ijms-21-02714-t003:** SNPs related to Efficacy in terms of Survival.

Gene	SNP	Results			Involved Candidate Drug *
		Genotype	HR	Association	
OS					
*MTHFR*	rs1801133	TC	0.65	↑ OS	Platinum-compounds Cyclophosphamide
*Unidentified*	rs7186128	GG	0.89	↑ OS	
*VDR*	rs1544410	GA	1.39	↓ OS	Etoposide
EFS					
*ABCB1*	rs2032582	GA	0.48	↑ EFS	Cyclophosphamide
*SOD2*	rs4880	TC	0.72	↑ EFS	
*NR1I2*	rs3814058	TT	0.62	↑ EFS	Etoposide
*ABCC1*	rs45511401	GT	1.79	↓ EFS	
*VDR*	rs1544410	GA	1.75	↓ EFS	
*Unidentified*	rs6539870	GG	1.61	↓ EFS	

EFS: event-free survival; HR: Hazard Ratio; OS: overall survival; SNP: Single Nucleotide Polymorphism. * According to literature and PharmGKB. *N* = 95 patients. ↑ means “increase” and ↓ “decrease”.

**Table 4 ijms-21-02714-t004:** SNPs (*n* = 96) evaluated in the study cohort and genes where they are located.

Genes	SNPs	Genes	SNPs
*ABCB1*	rs1045642	*FCGR3A*	rs396991
	rs1128503	*GSTA1*	rs3957357
	rs2032582	*GSTP1*	rs1695
	rs4148737	*MAD1L1*	rs1801368
	rs10276036	*MAP3K1*	rs726501
*ABCC1*	rs45511401	*MTHFR*	rs1801131
*ABCC2*	rs2273697		rs1801133
	rs3740066	*MTR*	rs1805087
	rs8187710	*NCF4*	rs1883112
	rs17222723	*NOS3*	rs1799983
*ABCC3*	rs4148416		rs2070744
*ABCC4*	rs9561778	*NQO1*	rs1800566
	rs16950650	*NQO2*	rs1143684
*ABCG2*	rs2231137	*NR1L2*	rs2276707
	rs2231142		rs3814058
*AKT1*	rs1130214	*OPMR1*	rs544093
	rs2494752	*RAC2*	rs13058338
*ALDH1A1*	rs6151031	*SEMA3C*	rs7779029
*BAIAP3*	rs9597	*SHMT1*	rs1979277
*BCL2*	rs2849380	*SLCO1B1*	rs2306283
*C8orf34*	rs1517114		rs4149015
*CBR1*	rs9024		rs4149056
	rs20572	*SLC19A1*	rs12659
*CBR3*	rs1056892		rs1051266
	rs8133052		rs7851395
*COMT*	rs9332377	*SLC22A16*	rs6907567
*CYBA*	rs4673		rs714368
*CYP2B6*	rs2279343		rs723685
	rs3211371		rs12210538
	rs3745274	*SLC31A1*	rs7851395
	rs8192709	*SLIT1*	rs2784917
	rs12721655	*SOD2*	rs4880
*CYP2C19*	rs4244285	*SOX10*	rs139887
*CYP2E1*	rs2070676	*TOP1*	rs6072262
	rs6413432	*TP53*	rs1042522
*CYP3A4*	rs2740574	*TPMT*	rs12201199
*CYP3A5*	rs776746	*UGT1A1*	rs4124874
	rs10264272		rs4148323
	rs41303343	*UGT1A9*	rs3832043
*DCBLD1*	rs17574269	*VDR*	rs731236
*DSCAM*	rs9981861		rs7975232
*DYNC2H1*	rs716274		rs1544410
*EGFR*	rs121434568	*VEGFA*	rs2010963
*EIF3A*	rs3740556	*XPC*	rs2228001
*ERCC1*	rs11615	*XRCC1*	rs25487
	rs3212986	*Unidentified*	rs879207
*ERCC2*	rs13181		rs6539870
	rs1799793		rs7186128
*FCGR2A*	rs1801274

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
