# Peer review of "MTHFR and VDR Polymorphisms Improve the Prognostic Value of MYCN Status on Overall Survival in Neuroblastoma Patients"

_ijms, 2020, doi:10.3390/ijms21082714_

Round 1

Reviewer 1 Report

In the present study, the authors performed a pharmacogenomics study in neuroblastoma patients. They found several SNP variants associated with treatment response, and prognosis. In particular, it was demonstrated that MTHFR rs1801133 and VDR rs1544410 provide a significant substratification in patients without MYCN amplification. These findings are very informative, providing important clues for treatment planning for NB patients. The following points should be addressed for the improvement of the manuscript.

  1. The authors should examine SNPs associated with high risk patients (either metastasis or MYCN amplification).
  1. Figure 2 and 3. The authors mentioned that significant p values were observed. However, it is unclear what were compared, is it in normal MYCN patients? in amplified ones?
  1. In comparison with the global, the OS rates of MTHFR rs1801133 TT and VDR rs1544410 GG were apparently different between normal MYCN and amplified MYCN. Is that just due to the number of patients? If the authors have other ideas, please discuss them.
  1. Introduction: Several sentences are related to materials and methods. They should be moved to the Materials and Methods section.
  1. Table 2 is missing.
  1. line 183. Supplementary Data Figure A2 → A1
  1. lines 367-368. The gene NR1I2 encodes PXR, but not VDR.
  1. line 410. homeocysteine → homocysteine

Author Response

Answers to Reviewer 1
We do thank the reviewer for the comments that will definitely improve the quality of the manuscript.
1. The authors should examine SNPs associated with high risk patients (either metastasis or MYCN amplification)
Although this analysis was not included in the original aims of this manuscript, we understand the concerns of the reviewer, so we have performed with our biostatistician an Elastic Net Logistic Regression trying to find if any of the SNPs in the panel were associated with High Risk patients. We had a total of 20 Non-High Risk and 75 High Risk patients. The only SNPs found to be associated with High Risk was rs1979277, with GG patients showing lower risk of being High Risk NB (P=0.016).
We are now including the following paragraph in Discussion section, line 306:
“Another relevant concern to assess before starting analyzing toxicity or efficacy was evaluating if there were SNPs in our panel that could be directly linked to a higher probability for a patient to be High Risk. In our whole cohort, 75 patients were High Risk and 20 non-High Risk. Only GG patients at SNP rs1979277 showed lower basal risk of being High Risk NB (p=0.016, Elastic Net logistic regression). This SNP is not part of any of the following results that correlate SNPs with efficacy or toxicity of the treatment, so we can conclude that no interference is caused. From another perspective, trying to find out which could be the explanation of this finding, the underlying mechanisms and the potential utility could be of interest for future researches”.
2. Figure 2 and 3. The authors mentioned that significant p values were observed. However, it is unclear what were compared, is it in normal MYCN patients? in amplified ones?
The reviewer is right, and we agree that this point should be better explained. We are adding now the corresponding clarifications in sentences at Figure 2 and 3 legends, and in lines 215, 222-224 in order to state to what comparisons the p values stand for. We also add a clearer explanation at Results section, line 189:
“In these two figures the analyses test if there are differences on the survival curves between the effect of MYCN status on its own (grey lines), compared to the effect of the corresponding MYCN status but dividing the patients according to their genotype for each SNP (blue or red lines). Significant p-values would mean that the stratification adding the genotype to MYCN status is statistically different from MYCN status alone, and the p-value will be particularly related to the most different variant comparing its colored line to the grey line”.
3. In comparison with the global, the OS rates of MTHFR rs1801133 TT and VDR rs1544410 GG were apparently different between normal MYCN and amplified MYCN. Is that just due to the number of patients? If the authors have other ideas, please discuss them.
According to the reviewer’s suggestion, we are now including further discussion about these results, in Discussion section, line 440:
“It is noticeable that amongst all the panels in Figures 2 and 3, considering only those with statistically significant p values, the results from MTHFR and VDR SNPs show apparent differences between normal and amplified MYCN. Regarding MTHFR and OS, while TC effect seems clear, the effects of TT and CC look opposite between normal or amplified MYCN patients. As to VDR, in both OS and EFS figures, the effect of GA is maintained in both scenarios, while GG and AA show different results. Unfortunately, we do not have an explanation for these findings. The number of patients in each genotype category could be responsible of these effects. Also, the treatment applied to the patients in each of the scenarios, normal vs. amplified, could be influencing these differences: the 66 MYCN amplified-patients are High Risk, so more intensively treated, while in the 29 normal MYCN patients, 9 of them are High Risk, and 20 are not (we have a total of 75 High Risk patients). Further studies are needed to explain the differences observed in those genotypes”.
4. Introduction: Several sentences are related to materials and methods. They should be moved to the Materials and Methods section.
According to the reviewer’s suggestion, we have deleted the following sentences form Introduction Section, and assured that the information is present in Materials and Methods section
Deleted: Our panel was finally designed including 96 SNPs, belonging to a total of 60 different genes. A total of 95 NB patients, treated at the Pediatric Oncology Unit of Hospital La Fe (Valencia, Spain) with available frozen peripheral blood samples collected after written informed consent obtention, were included in this study. This project was approved by Hospital La Fe Ethics Committee.
5. Table 2 is missing
Sorry for the mistake. This has been corrected and the following tables renamed, too.
6. line 183. Supplementary Data Figure A2 → A1
Sorry again for the mistake, it has been corrected.
7. lines 367-368. The gene NR1I2 encodes PXR, but not VDR
We have double checked this point: NR1I1 is the name of the gene that we wrote, and it is an alternative name for VDR gene. We were not talking about NR1I2, which, as the reviewer said, is an alternative name for PXR gene.
8. line 410. homeocysteine → homocysteine
We thank the reviewer for the observation. We have changed the spelling.

Reviewer 2 Report

The manuscript titled ‘MTHFR and VDR polymorphisms improve the prognostic value of MYCN status on Overall Survival in neuroblastoma patients’ is well written. There are few comments that need to be addressed: 

  1. Introduction is too long. Authors may consider removing the results paragraph (lines 105-114) from the introduction section or just express the results in one sentence.
  2. Figure 1 is unclear. Please organize and present the info in a different way.
  3. As such, the manuscript is very long and author should attempt to streamline the discussion and highlight the key points. Mechanisms of link between MTHFR and VDR SNPs and NB are discussed in a non-specific way, more specific elaboration is needed.

Author Response

Answers to Reviewer 2

We really thank the reviewer for the comments. We do think that the manuscript has increased its quality after their implementation.

1-Introduction is too long. Authors may consider removing the results paragraph (lines 105-114) from the introduction section or just express the results in one sentence.

We thank the reviewer for the comments, and according to the suggestion, we have summarized the main results in one sentence, at the end of Introduction section, line 100.

“The main results found: 1) frequencies different from expected in 19 SNPs; 2) 8 SNPs related with severe or life-threatening chemotherapy-associated toxicities; 3) SNPs influencing RIT in patients’ with high-risk NB and 4) SNPs in MTHFR, VDR and SOD2 genes identified as prognostic factors of OS or EFS, independently of MYCN amplification, underlying differential survival prediction possibilities independent of MYCN status”.

Also, Introduction had been already shortened according to suggestion from Reviewer 1. The following paragraph was removed:

“Our panel was finally designed including 96 SNPs, belonging to a total of 60 different genes. A total of 95 NB patients, treated at the Pediatric Oncology Unit of Hospital La Fe (Valencia, Spain) with available frozen peripheral blood samples collected after written informed consent obtention, were included in this study. This project was approved by Hospital La Fe Ethics Committee”.

2-Figure 1 is unclear. Please organize and present the info in a different way.

According to the reviewer’s suggestion, we have remade Figure 1 and we have clarified the explanation, introducing a new paragraph in the text, line 120:

“Figure 1 shows the number of samples included in the whole study, indicating the subsets of samples that have been chosen for the different analyses reported in Results section. From the total amount of 100 newly diagnosed frozen samples that were available in the retrospective collection, 5 were not suitable for genotyping, so 95 remained for use. Those 95 were considered as the whole cohort. Each of the following paragraphs in this section explains the criteria for choosing the subgroups. In brief, the SNP variants frequency analyses, OS and EFS studies were performed on the whole cohort. From those 95 samples, only 41 were High Risk NB treated with Rapid COJEC and on those, RIT relation to SNPs analysis was analyzed. Inside that group of 41 patients, 35 completed the whole Induction treatment in the same hospital, so analysis of chemotherapy-toxicity related to SNP could be performed”

3-As such, the manuscript is very long and author should attempt to streamline the discussion and highlight the key points. Mechanisms of link between MTHFR and VDR SNPs and NB are discussed in a non-specific way, more specific elaboration is needed.

According to the reviewer’s suggestions for this point, we have performed several changes:

-We have deleted a whole paragraph from Discussion section:

The results of this work are the associations between the panel of SNPs and the treatment outcomes in the patients, in terms of efficacy and toxicity, and can be divided in three parts: 1) frequencies of SNP variants analyses (95 patients); 2) SNPs found to be related with severe toxicity during Induction treatment (35 patients, High Risk, all with Rapid COJEC completed at La Fe Hospital); 3) SNPs found to be related with efficacy of the treatment in terms of 3.1) Response to Induction Therapy, in 41 High Risk patients treated with Rapid COJEC, with the currently used criteria for results interpretation (mCR vs non mCR); 3.2) Efficacy in terms of OS and EFS, in the whole cohort: 95 patients.

-We have tried to improve several sentences, as in line 380: “We hypothesize that this could be the link for the effects observed: lower levels of VDR could results in lower levels of etoposide metabolism and therefore, lower RIT”.

And 426: “We could hypothesize that an increase in MYCN expression, could lead to downregulation of MTHFR and this subsequently, lead to a decrease in the routes of DNA synthesis and DNA methylation. If this is true, then the variant TT could be boosting even more the effect in MYCN amplified patients, since TT has been described as lowering the activity of MTHFR enzyme, and that is what we see in our patients”.

-As the reviewer suggest, in agreement also with Reviewer 1, we have further discussed MTHFR and VDR results, by adding a new paragraph in line 440:

It is noticeable that amongst all the panels in Figures 2 and 3, considering only those with statistically significant p values, the results from MTHFR and VDR SNPs show apparent differences between normal and amplified MYCN. Regarding MTHFR and OS, while TC effect seems clear, the effects of TT and CC look opposite between normal or amplified MYCN patients. As to VDR, in both OS and EFS figures, the effect of GA is maintained in both scenarios, while GG and AA show different results. Unfortunately, we do not have an explanation for these findings. The number of patients in each genotype category could be responsible of these effects. Also, the treatment applied to the patients in each of the scenarios, normal vs. amplified, could be influencing these differences: the 66 MYCN amplified-patients are High Risk, so more intensively treated, while in the 29 normal MYCN-patients, 9 of them are High Risk, and 20 are not (we have a total of 75 High Risk patients). Further studies are needed to explain the differences observed in those genotypes”.

Reviewer 3 Report

Review of an article “MTHFR and VDR polymorphisms improve the prognostic value of MYCN status on Overall Survival in neuroblastoma patients” by Gladys Oliver and coauthors submitted to MDPI.

It is vital to know polymorphic variations in genes contributing to human diseases. Polymorphisms may affect the course of a disease and affect the outcomes of the chemotherapy treatment. Information about these effects helps physicians to maximize the efficacy of the treatment and minimize the toxicity of medication. Oliver and colleagues investigated the impact of chemotherapy-related single nucleotide polymorphic variations (SNPs) on the efficacy of neuroblastoma treatment and the toxicity of certain medications. This is an important area of investigation, and the results will be interesting for the IJMS readers.

The following correction should be done.

Abstract

Lines 24- 28 It is a long and hard to read sentence should be cut into shorter and more comfortable to read and understand sentences.

Lines 28-29 “SNP “profile” was associated with grade 3-4 gastrointestinal and infectious toxicities, with rs2228001, rs7779029, rs1799793, rs1801133, rs2306283, rs2784917, rs6907567 and rs7186128”. The sense of the sentence is unclear. Sjould be rewritten.  

Lines 22-37 Abstract overall is overloaded with abbreviations, digits, names of the SNPs, etc., and so it is hard to read. Should be rewritten to become more “reader-friendly”.

Introduction

Lines 105-114. At the end of Introduction, the authors should present a SHORT summary of the results (1-2 sentences) without detailed information about all the SNPs they studies (lines 29-31).  

Results

Lines 32-33 Figure 1 should be prepared more professionally

Table 2 is absent

Lines 186-187

“With the aim of going further, we combined the information by adding to the survival curves according to MYCN status the results of the SNPs selected to influence OS, in Figure 2 and EFS, in Figure 3.”

The sense of the sentence is unclear, should be rewritten.

Discussion

Lines 245-247

“The aim is to find better ways to predict which patients will be good or bad responders to the treatment, in order to use the findings (once confirmed) in the near future, to help identify “good/bad responders” at the earliest possible, according to the preemptive PGx testing”.

Hard to read and understand the sentence. Should be rewritten in a “more reader-friendly” way.

 Methods. 

Line 463. “DNA concentration and purity were measured with a NanoDrop spectrophotometer.”

The authors should explain how they measured DNA purity by spectrophotometry.

Author Response

Answers to Reviewer 3

We do thank the reviewer for the comments, which will definitely improve the quality of the manuscript

Abstract

Lines 24- 28 It is a long and hard to read sentence should be cut into shorter and more comfortable to read and understand sentences.

Lines 28-29 “SNP “profile” was associated with grade 3-4 gastrointestinal and infectious toxicities, with rs2228001, rs7779029, rs1799793, rs1801133, rs2306283, rs2784917, rs6907567 and rs7186128”. The sense of the sentence is unclear. Sjould be rewritten.  

Lines 22-37 Abstract overall is overloaded with abbreviations, digits, names of the SNPs, etc., and so it is hard to read. Should be rewritten to become more “reader-friendly”.

According to the reviewer’s suggestions, the whole Abstract has been modified, simplifying the sentences and excluding expendable digits and abbreviations.

Single nucleotide polymorphisms (SNPs) in Pharmacogenetics can play an important role in the outcomes of the chemotherapy treatment in Neuroblastoma, helping doctors maximize efficacy and minimize toxicity. Employing AgenaBioscience MassArray, 96 SNPs were genotyped in 95 patients looking for associations of SNP with response to induction therapy (RIT) and grade 3-4 toxicities, in High Risk patients. Associations of SNPs with overall (OS) and event-free (EFS) survival in the whole cohort were also explored. Cox and logistic regression models with Elastic net penalty were employed. Association with grade 3-4 gastrointestinal and infectious toxicities was found for 8 different SNPs. Better RIT was correlated with rs726501 AG, rs3740066 GG, rs2010963 GG and rs1143684 TT (OR = 2.87, 1.79, 1.23, 1.14, respectively). EFS was affected by to rs2032582, rs4880, rs3814058, rs45511401, rs1544410 and rs6539870. OS was influenced by rs 1801133, rs7186128 and rs1544410. Remarkably, rs1801133 in MTHFR (p=0.02) and rs1544410 in VDR (p=0.006) also added an important predictive value for OS to the MYCN status, with a more accurate substratification of the patients. Although validation studies in independent cohorts will be required, the data obtained supports the utility of Pharmacogenetics for predicting Neuroblastoma treatment outcomes.

Introduction

Lines 105-114. At the end of Introduction, the authors should present a SHORT summary of the results (1-2 sentences) without detailed information about all the SNPs they studies (lines 29-31).  

According to the reviewer’s suggestion, and also in line with Reviewer 2, we have changed and shortened that paragraph, line 100:

“The main results found: 1) frequencies different from expected in 19 SNPs; 2) 8 SNPs related with severe or life-threatening chemotherapy-associated toxicities; 3) SNPs influencing RIT in patients’ with high-risk NB and 4) SNPs in MTHFR, VDR and SOD2 genes identified as prognostic factors of OS or EFS, independently of MYCN amplification, underlying differential survival prediction possibilities independent of MYCN status”

Results

Lines 32-33 Figure 1 should be prepared more professionally

According to the suggestion, also in line with Reviewer 2, we have remade Figure 1 and also explained it with more detail in the text, line 120 and Figure 1 legend

Table 2 is absent

We are sorry for this mistake. It has been corrected and the rest of the tables renamed consequently

Lines 186-187

“With the aim of going further, we combined the information by adding to the survival curves according to MYCN status the results of the SNPs selected to influence OS, in Figure 2 and EFS, in Figure 3.”

The sense of the sentence is unclear, should be rewritten.

We agree with the reviewer, this is an important point in the manuscript and should be explained clearly. We have rephrased the sentence, line 186:

               “With the aim of going on a step further, we decided to check if the combination of these new findings (table 3) with MYCN status could improve the identification of patients with better or poorer prognosis in the survival curves. The results are shown in Figure 2 for OS and Figure 3 for EFS”.

And this is followed by a new paragraph, line 189, that Reviewer 1 also suggested:

               “In these two figures the analyses test if there are differences on the survival curves between the effect of MYCN status on its own (grey lines), compared to the effect of the corresponding MYCN status but dividing the patients according to their genotype for each SNP (blue or red lines). Significant p-values would mean that the stratification adding the genotype to MYCN status is statistically different from MYCN status alone, and the p-value will be particularly related to the most different variant comparing its colored line to the grey line”.

Discussion

Lines 245-247

“The aim is to find better ways to predict which patients will be good or bad responders to the treatment, in order to use the findings (once confirmed) in the near future, to help identify “good/bad responders” at the earliest possible, according to the preemptive PGx testing”.

Hard to read and understand the sentence. Should be rewritten in a “more reader-friendly” way.

According to the reviewer, we have rephrased the sentence in line 255:

               “The aim is to find better ways to predict which patients will be “good or bad responders” to the treatment. If we employ preemptive PGx testing for this, we would be able to identify these patients at the earliest possible and adjust their treatment in a personalized way”.

 Methods. 

Line 463. “DNA concentration and purity were measured with a NanoDrop spectrophotometer.”

The authors should explain how they measured DNA purity by spectrophotometry.

We thank the reviewer for the appreciation, and we have modified the sentence in line 487 accordingly:

“DNA concentration and purity were measured with a NanoDrop spectrophotometer (NanoDrop Technologies Inc., Wilmington, DE, USA), according to the manufacturer’s instruction, using the provided absorbance at 260nm for the first and the 260/280 ratio for the second”.

Round 2

Reviewer 1 Report

The authors had adequately and sincerely addressed my concerns.

This manuscript has been improved.

Reviewer 2 Report

The comments were appropriately addressed.